# FAILURE MAKES THE AGENT STRONGER: ENHANCING ACCURACY THROUGH STRUCTURED REFLECTION FOR RELIABLE TOOL INTERACTIONS

## ABSTRACT

Tool-augmented large language models (LLMs) are typically trained via supervised imitation learning or coarse-grained reinforcement learning, approaches that primarily optimize one-shot tool calls. Existing practices of self-reflection largely rely on heuristic prompting or unidirectional reasoning traces: the model is encouraged to "think more," rather than to treat error diagnosis and correction as a learnable capability. This makes them fragile in multi-turn interaction settings—once a call fails, the model tends to repeat the same mistake instead of recovering. To address this issue, we propose structured reflection, which transforms the "from error to repair" process into a first-class, controllable, and trainable action. The agent produces a concise yet precise reflection process: specifically, the model diagnoses the error based on evidence from the previous step and then proposes a correct and executable follow-up call. During training, we combine DAPO and GSPO's objective functions and design a more principled reward mechanism tailored to tool calling, optimizing the stepwise strategy Reflect → Call → Final. To evaluate this capability, we introduce Tool-Reflection-Bench, a lightweight benchmark dataset that programmatically verifies structural validity, executability, parameter correctness, and result consistency. Tasks in the benchmark are constructed as miniature trajectories of Erroneous Call → Reflection → Corrected Call and are split into disjoint training and testing sets. Experiments on BFCL v3 and Tool-Reflection-Bench show that our method achieves significant improvements in multi-turn tool-call success rates and error recovery, while also reducing redundant calls. These results demonstrate that making reflection explicit and treating it as an optimization objective can substantially enhance the reliability of tool interaction, providing a reproducible pathway for agents to grow stronger by learning from failure. We will release all the code and datasets as open source once the paper is accepted by the community.

## 1 INTRODUCTION

The integration of external tools with large language models through tool calling represents a significant breakthrough in the development of agents. It transforms large language models from mere text generators into highly practical tools for interacting with humans WANG et al. (2025); Qu et al. (2024a), significantly enhancing the ability of AI agents to solve complex real-world tasks Huang et al. (2024); Qin et al. (2023); Qu et al. (2024b). Tool calling bridges the gap between the vast internal knowledge of LLMs and external resources, enabling LLMs to access up-to-date information, perform delicate computations, and more, thereby unlocking their broad potential for applications across multiple domains Zhong et al. (2023); Theuma & Shareghi (2024); Hao et al. (2024).

Currently, the training of tool-call capabilities in large language models typically relies on supervised fine-tuning and reinforcement learning Chen et al. (2025b); Qian et al. (2025), where these methods optimize the ability for single-turn tool calls through carefully designed reward mechanisms. However, these approaches face several challenges in the context of tool calling. First, the issue of rewards in tool calling is particularly prominent—small errors in parameter selection or formatting often render the entire function call invalid, thus limiting the effective learning signal

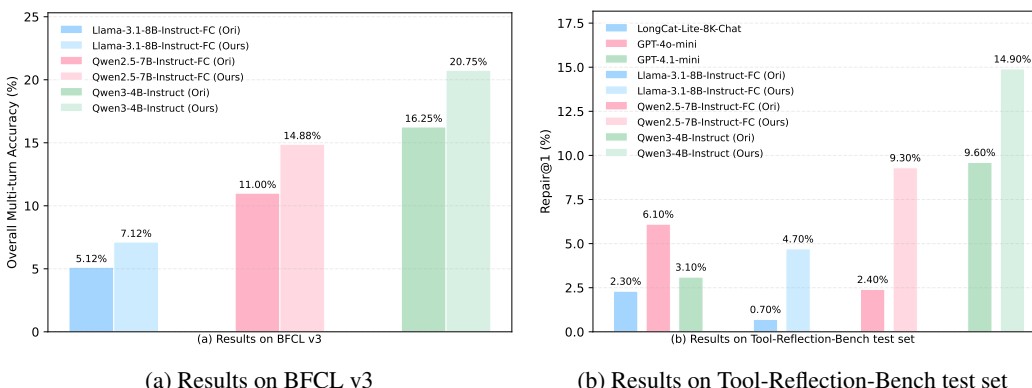

(a) Results on BFCL v3      (b) Results on Tool-Reflection-Bench test set

Figure 1: In the experiments on BFCL v3 and Tool-Reflection-Bench, our method significantly improves the multi-turn tool-calling accuracy of several open-source LLMs on BFCL v3. At the same time, it substantially enhances the error-repair rate for tool calls on the Tool-Reflection-Bench test set, achieving performance that even surpasses that of closed-source LLMs with comparable parameter sizes.

Lattimer et al. (2024). Second, existing methods generally rely on unidirectional reasoning, which, while sufficient for simpler scenarios, has clear limitations: when LLMs make mistakes during tool calls, they often struggle to locate the root cause of the error Li et al. (2025). While generating correct function calls is crucial, it is even more important for LLMs to learn how to identify and correct their own mistakes Ye et al. (2024).

To address the above-mentioned issues, we propose an innovative reflection process aimed at error localization and correction through explicit reflection steps, which differs from existing forward reasoning methods. Specifically, we design a process in which the LLM intentionally makes mistakes during tool calls, carefully crafts reflection content based on the errors, and then generates the correct call. Through this approach, we transform the self-correction ability of large models from a heuristic process Yang et al. (2024) into a clear, trainable capability. Our training approach is primarily reinforcement learning–based. During the reinforcement learning process, we specifically design a customized reward mechanism tailored for tool-calling scenarios, with a particular emphasis on multi-turn interactions. Concretely, the reward design encompasses multiple dimensions, including format reward, tool-name reward, parameter reward, and semantic reward of reflection, which together provide the model with multi-dimensional feedback and effectively guide its learning, and we further combine DAPO's decoupled clipping range and dynamic sampling—expanding exploration while skipping near-zero-advantage rollouts—with GSPO's sequence-level importance sampling and same-granularity clipping, which avoids token/sequence mismatch and stabilizes optimization. With this training methodology, our approach equips LLMs with genuine self-reflection and error-correction capabilities. On the BFCL v3 benchmark, our method yields significant improvements in LLM accuracy for multi-turn tool calling, thereby demonstrating its effectiveness in real-world applications.

We construct a Tool-Reflection-Bench based on the BUTTON dataset Chen et al. (2024) style. First, we collected tool-call failure cases from real-world scenarios and various benchmarks, analyzing and summarizing several common failure patterns. Next, We selected several existing tool-call datasets Qin et al. (2023); Liu et al. (2024b) and randomly combined them according to the call style of the BUTTON dataset and introduced these failure patterns into the data, disrupting the originally correct call processes to generate failure cases. Finally, we meticulously designed a reflection process to repair these failures, resulting in successful tool calls. The training set includes the complete process described above to train LLMs to achieve true self-correction capabilities, while the test set only contains the first two steps, used to evaluate the self-correction abilities of the LLMs. By constructing the Tool-Reflection-Bench in this manner, combined with our custom reward mechanism for tool calling, we have made breakthroughs in LLMs' self-correction abilities during training. Particularly in multi-turn tool-calling scenarios, we observed significant improvements in accuracy. Through the reasoning process from failure to correction, LLMs can more effectively identify and learn from

potential mistakes, thus enhancing the model's stability and robustness in interactions. This makes the agent's behavior more robust and powerful.

In summary, our contributions are as follows:

- We introduce an explicit, trainable reflection process that diagnoses the cause of a failed tool call using prior evidence and proposes a corrected, executable call. This transforms the "from failure to repair" process from a heuristic method into a learnable action strategy, enabling LLMs to genuinely possess self-reflection and error-correction capabilities, thereby enhancing the agent's multi-turn interactions with users.

- We design a more effective reward mechanism for tool call, tailored for RL training, using a GRPO-style objective function. This approach employs multi-dimensional rewards—format executability, tool name accuracy, parameter correctness, and semantic consistency—to mitigate sparse rewards and propagate signals across multi-turn trajectories.

- We propose Tool-Reflection-Bench, which collects failure patterns from real interaction scenarios and benchmark datasets, injects perturbations into correct calls, and attaches a reflection process to repair the calls. This allows for training LLMs in their Self-Correction ability in tool-calling scenarios.

- Our method significantly improves the accuracy of multi-turn tool calls and the ability to recover from tool call errors, while maintaining competitive single-turn tool call performance. We validate this by experiments on BFCL v3 Patil et al. and Tool-Reflection-Bench.

## 2 RELATED WORKS

### 2.1 TOOL-AUGMENTED LARGE LANGUAGE MODELS

Integrating external tools into large language models has become a key approach to enhancing their functionality, surpassing the simple task of text generation. Traditional LLMs are limited by static knowledge, constrained to the data they were trained on. However, tool-augmented models extend the capabilities of LLMs by enabling them to interact with external resources Zhang et al. (2024); Hao et al. (2025) (such as APIs Li et al. (2023), databases, and computational engines) through tool calls. This extension allows LLMs to access real-time data, perform external computations, and even interface with external hardware, making them more practical for solving complex real-world tasks that require dynamic information or specific external operations Chen et al. (2025a). ToolBench Qin et al. (2023) demonstrates the feasibility of integrating external tool calls into LLMs. Through such systems, LLMs can handle more specialized tasks. However, one major challenge of tool augmentation is how to effectively train LLMs to use these tools. Existing training methods, such as supervised fine-tuning and reinforcement learning, typically focus on optimizing single tool calls. This type of interaction often does not involve multi-turn tool calls or responses, which makes the limitations of current methods particularly apparent when errors occur during tool usage. In such cases, the model's ability to recover from errors becomes crucial.

### 2.2 SELF-CORRECTION IN LLMS

Self-correction in large language models refers to the model's ability to diagnose its own errors and correct them based on previous actions Huang et al. (2023); Liu et al. (2024a). However, this area has not been fully explored. Existing self-correction techniques mostly rely on heuristic methods or unidirectional reasoning processes Renze & Guven (2024).

Self-Refine framework Madaan et al. (2023), which involves having LLMs provide an initial response, followed by a reflection process where the model identifies flaws and makes improvements. Specifically, the same LLM acts as both the responder and the evaluator: the model first generates an initial response, then self-reflects and iteratively revises the output. This approach has been shown to enhance the performance of LLMs in certain domains. However, subsequent studies Wu et al. (2024); Vladika et al. (2025) have found that relying solely on the model itself often fails to detect subtle errors. Some research Jiang et al. (2025); Zhao et al. (2025a) has introduced auxiliary verifiers (such as additional models or mechanisms Saveliev & Dendiuk (2024); Feng et al. (2025)) to help check the correctness of the initial response. This external self-checking assistance avoids

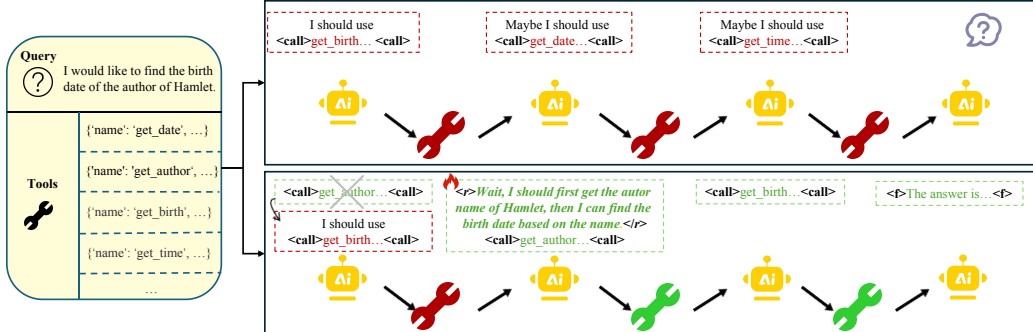

Figure 2: We illustrate the effectiveness of our method with an example. As shown in the figure, the left side presents the tool panel, while the upper-right part depicts industry-standard self-correction approaches, where models attempt to fix errors through heuristic trial-and-error reasoning or by relying on external feedback. In contrast, the lower-right part shows our approach: we introduce an explicit forced reflection process <r>, enabling the model to truly master the ability to repair errors based on its own failures.

unnecessary repeated revisions, improving efficiency and enhancing the model's reasoning and verification capabilities. However, this approach remains highly sensitive to the specific phrasing of the prompts, with different prompt wordings leading to varying results Liu et al. (2024a).

However, even though these methods have somewhat improved the capabilities of LLMs, their essence still relies on external feedback to assist in correcting their own errors. In contrast, our work introduces a reflection method that transforms self-correction into a trainable and controllable capability. The reflection process is an explicit part of the task, where the model actively evaluates its previous actions, identifies errors, and generates explicit corrections. This process is achieved through error localization, diagnosis, and correction, which goes beyond simple unidirectional reasoning and can be integrated into the training process. By providing supervisory signals during training, our approach enables LLMs to truly possess self-correction capabilities, which are then reflected in tool calling tasks.

## 3 METHOD

### 3.1 TOOL-REFLECTION-BENCH

The construction of Tool-Reflection-Bench consists of the following steps: perturbation-based disruptions, positive samples transformations, and the reflection repair process. The original positive samples are derived from BUTTON Chen et al. (2024) transformations and self-constructed based on few-shot prompts. The entire benchmark is divided into a training set and a test set, with approximately 5,000 samples in the training set, in addition to the reflection-augmented data constructed as described above, the training set also contains a very small portion of original data drawn from BUTTON Chen et al. (2024) and XLAM Zhang et al. (2024). And around 1,000 samples in the test set, the test set is exclusively composed of perturbation-derived items and does not include raw, unperturbed positives from BUTTON or XLAM.

#### 3.1.1 PERTURBATION-BASED DISRUPTIONS

Let the initial correct message sequence be

$$D^+ = \left( m_0^{\text{sys}}, m_1^{\text{usr}}, m_2^{\text{ast}}, m_3^{\text{tool}}, m_4^{\text{ast}}, m_5^{\text{tool}}, \ldots, m_{2k}^{\text{ast}}, m_{2k+1}^{\text{tool}}, \ldots, m_n^{\text{final}} \right), \quad (1)$$

where $m_0^{\text{sys}}$ is the system prompt, $m_1^{\text{usr}}$ the user query, $m_{2i}^{\text{ast}}$ the assistant's $i$-th tool call in structured form (e.g., `<call>[{...},{...},...]</call>`), $m_{2i+1}^{\text{tool}}$ the tool return (JSON), and $m_n^{\text{final}}$ the final answer.

We define a set of disruption operators

$$\mathcal{P} = \{P_1, P_2, P_3, P_4\}, \tag{2}$$

each operating on an assistant call $m_{2k}^{\text{ast}}$ and instantiating a common failure mode:

1. $P_1$ **call-order swap**: replace the current tool call with the next-round tool call dialogue and force an error.

2. $P_2$ **redundant call**: repeat the same tool at the step (unchanged/irrelevant arguments) and force an error.

3. $P_3$ **missing call**: replace the intended tool by another tool and force an error.

4. $P_4$ **argument error**: randomly corrupt the arguments of a call (missing/typed/alias/boundary) and force an error.

These operators specify how a correct tool call can be broken.

### 3.1.2 POSITIVE SAMPLES TRANSFORMATIONS

Given a clean trajectory $D^+$ and a chosen operator $P_j \in \mathcal{P}$ acting on step $2k$, we produce the negative (erroneous) context; no repair is performed in this step. We construct the erroneous call

$$\tilde{m}_{2k}^{\text{ast}} = \text{ApplyPerturbation}\big(m_{2k}^{\text{ast}}, P_j\big), \tag{3}$$

and simulate the tool's error feedback with a LLM $\mathcal{L}$:

$$\tilde{m}_{2k+1}^{\text{tool}} = \mathcal{L}\big(\tilde{m}_{2k}^{\text{ast}}; \mathcal{L}\big). \tag{4}$$

This yields the **negative** trajectory prefix

$$D^- = \text{Perturb}\big(D^+, P_j\big) = \Big(m_0^{\text{sys}}, m_1^{\text{usr}}, \ldots, \tilde{m}_{2k}^{\text{ast}}, \tilde{m}_{2k+1}^{\text{tool}}\Big), \tag{5}$$

which will later serve as evidence of failure. At this stage, the item consists only of the broken call and its error signal.

### 3.1.3 REFLECTION REPAIR PROCESS

Given a clean trajectory $D^+$ and its perturbed prefix $D^-$, we present the LLM with a paired view of the step-$2k$ evidence:

$$\text{clean: } (m_{2k}^{\text{ast}}, m_{2k+1}^{\text{tool}}) \quad \text{vs.} \quad \text{broken: } (\tilde{m}_{2k}^{\text{ast}}, \tilde{m}_{2k+1}^{\text{tool}}). \tag{6}$$

The model outputs a response.

$$\langle\text{reflect}\rangle r \langle/\text{reflect}\rangle, \tag{7}$$

where $r$ briefly diagnoses the discrepancy, and $c$ proposes the fixed tool call. We then apply human supervision to obtain $(r^\star, c^\star)$, with $c^\star$ set to the original correct call:

$$(r, c) \xRightarrow[\text{human supervision}]{\text{post–editing}} (r^\star, c^\star), \tag{8}$$

$$\mathcal{L}_\Sigma(c^\star) = \text{Success Call}. \tag{9}$$

The finalized item is packaged as

$$x = \big(D^-, r^\star, c^\star, D_{>2k+1}^+\big), \tag{10}$$

where $D_{>2k+1}^+$ is the untouched suffix of $D^+$ (including $m_n^{\text{final}}$). We retain $x$ only if: (i) tags/JSON are well-formed; (ii) $c^\star$ is executable; (iii) $r^\star$ correctly cites the clean–broken contrast.

## 3.2 REWARD DESIGN

**Preliminary.** Given a model completion $C$ and a ground truth $G$, we decompose both into three (possibly empty) parts:

$$C \mapsto \big(c_{\text{ref}}, C_{\text{calls}} = \{c_i\}_{i=1}^m, c_{\text{final}}\big), \qquad G \mapsto \big(g_{\text{ref}}, G_{\text{calls}} = \{g_j\}_{j=1}^n, g_{\text{final}}\big). \tag{11}$$

Here $c_{\text{ref}}$ (*reflection*) is the diagnosis text wrapped in <reflect></reflect>, $C_{\text{calls}}$ is the multiset of tool calls wrapped in <call></call>s produced by the model, and $c_{\text{final}}$ is the message wrapped in <final></final>. The ground truth can also be decomposed into three parts mentioned above.

**Component scores.** We compute three component scores:

$$s_{\text{ref}} = \text{Sim}(c_{\text{ref}}, g_{\text{ref}}), \qquad s_{\text{call}} = \mathbb{I}[\text{EqualCalls}(C_{\text{calls}}, G_{\text{calls}})], \qquad s_{\text{final}} = \text{Sim}(c_{\text{final}}, g_{\text{final}}), \quad (12)$$

where $\text{Sim} \in [0, 1]$ is a semantic similarity function, and $\mathbb{I}[\cdot]$ is the indicator:

$$\mathbb{I}[P] = \begin{cases} 1, & \text{if } P \text{ is true}, \\ 0, & \text{otherwise}. \end{cases} \quad (13)$$

We say $\text{EqualCalls}(C_{\text{calls}}, G_{\text{calls}})$ holds iff the two sets of produced calls can be put in a one-to-one correspondence such that for every matched pair the **tool name** is identical and the **argument** is identical.

**Normalization with presence masks.** Our goal is to keep the aggregated score in $[0, 1]$ even when an instance specifies only a subset of targets (e.g., only `<call>` without `<reflect>` or `<final>`). To this end we use normalization to renormalize over the parts that actually appear in the ground truth, so the maximum remains 1 regardless of how many parts are present.

We define

$$I_{\text{r}} = \mathbb{I}[g_{\text{ref}} \neq \varnothing], \qquad I_{\text{c}} = \mathbb{I}[|G_{\text{calls}}| > 0], \qquad I_{\text{f}} = \mathbb{I}[g_{\text{final}} \neq \varnothing]. \quad (14)$$

Let $(w_{\text{r}}, w_{\text{c}}, w_{\text{f}}) \geq 0$ be *normalized* base weights (e.g., $w_{\text{r}} + w_{\text{c}} + w_{\text{f}} = 1$). We renormalize over the active parts via

$$W_{\text{act}} = w_{\text{r}} I_{\text{r}} + w_{\text{c}} I_{\text{c}} + w_{\text{f}} I_{\text{f}}. \quad (15)$$

The aggregated structure/semantics score is then

$$S = \frac{w_{\text{r}} I_{\text{r}} s_{\text{ref}} + w_{\text{c}} I_{\text{c}} s_{\text{call}} + w_{\text{f}} I_{\text{f}} s_{\text{final}}}{W_{\text{act}}}. \quad (16)$$

This normalization yields a **consistent** scoring standard across fully and partially supervised instances, avoiding artificial deflation of scores when some targets are absent.

**Format/penalty factor.** We designed structural penalties tailored for tool-call data formats. Specifically, $P_{miss}$ accounts for cases where the tool is not invoked at all, while $P_{extra}$ and $P_{count}$ penalize redundant calls and mismatches in the total number of calls, respectively. Let

$$n = |G_{\text{calls}}|, \qquad m = |C_{\text{calls}}|, \quad (17)$$

Here $n$ and $m$ denote the number of tools invoked in the ground truth and completion calls. Define the three components:

$$P_{\text{miss}} = w_{\text{ref}} \mathbb{I}[g_{\text{ref}} \neq \varnothing \wedge c_{\text{ref}} = \varnothing] + w_{\text{final}} \mathbb{I}[g_{\text{final}} \neq \varnothing \wedge c_{\text{final}} = \varnothing] + w_{\text{calls}} \mathbb{I}[n > 0 \wedge m = 0], \quad (18)$$

$$P_{\text{extra}} = w_{\text{ref}} \mathbb{I}[c_{\text{ref}} \neq \varnothing \wedge g_{\text{ref}} = \varnothing] + w_{\text{final}} \mathbb{I}[c_{\text{final}} \neq \varnothing \wedge g_{\text{final}} = \varnothing] + w_{\text{calls}} \mathbb{I}[m > 0 \wedge n = 0], \quad (19)$$

$$P_{\text{count}} = w_{\text{calls}} \mathbb{I}[n > 0 \wedge m > 0 \wedge n \neq m] \frac{|n - m|}{\max(n, m)}. \quad (20)$$

Let EqualCalls be the schema-strict equality on bags of calls. We use a reduction factor

$$r = \begin{cases} r_{\text{reduce}}, & \text{if } \text{EqualCalls}(C_{\text{calls}}, G_{\text{calls}}), \\ 1, & \text{otherwise}, \end{cases} \qquad r_{\text{reduce}} \in (0, 1]. \quad (21)$$

Aggregate the penalty as

$$P_{\text{total}} = P_{\text{miss}} + \beta_{\text{extra}} P_{\text{extra}} + \gamma_{\text{count}} P_{\text{count}}, \quad (22)$$

and define the instance-wise format factor

$$\text{FormatFactor}(C, G) = \begin{cases} 1, \text{if } P_{\text{miss}} = 0 \wedge P_{\text{extra}} = 0 \wedge P_{\text{count}} = 0, \\ \max(0, \min(1, 1 - \lambda_m P_{\text{total}} r)), \text{otherwise}. \end{cases} \quad (23)$$

Here $\beta_{\text{extra}}, \gamma_{\text{count}}, \lambda_m \geq 0$ control the strength of extra-part, count-mismatch, and overall scaling penalties, respectively; $(w_{\text{ref}}, w_{\text{calls}}, w_{\text{final}}) \geq 0$ are part weights.

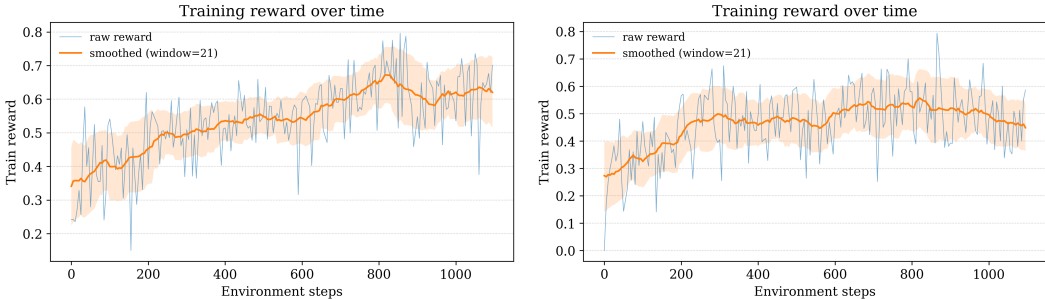

(a) The reward curve of llama-3.1-8b-Instruct during RL training

(b) The reward curve of qwen2.5-7b-Instruct during RL training

Figure 3: The reward curves of llama-3.1-8B and Qwen2.5-7B during training, showing an overall upward trend.

**Core reward and backoff.** The core reward is

$$R_{\text{core}} = S \cdot F. \tag{24}$$

Early in training, $S$ contains a binary component ($s_{\text{call}} \in \{0, 1\}$) and $F$ applies hard penalties; small formatting or argument errors can drive $R_{\text{core}}$ close to zero. This yields sparse or unstable gradients and large variance across samples. To stabilize learning and provide a dense shaping signal when the exact-match objective is not yet achieved, we introduce a similarity backoff:

$$R_{\text{total}} = \begin{cases} \text{clip}_{[0,1]}(R_{\text{core}}), & R_{\text{core}} \geq \varepsilon, \\ \text{clip}_{[0,1]}\big(w_{\text{b}} \cdot \text{Sim}(\text{concat}(C), \text{concat}(G))\big), & \text{otherwise}, \end{cases} \tag{25}$$

where $w_{\text{b}} \in (0, 1]$ and $\text{concat}(\cdot)$ linearizes the messages. We use $\text{clip}_{[0,1]}(x) = \max(0, \min(1, x))$ to keep rewards bounded.

### 3.3 RL FOR TOOL-REFLECTION-BENCH

We adopt a reinforcement-learning objective for tool calling that combines two complementary ideas: **(i) DAPO-style decoupled clipping** Yu et al. (2025): we use a decoupled clipping range with different lower/upper bounds ($\varepsilon_{\text{low}}, \varepsilon_{\text{high}}$) and a clip-higher policy (a looser upper bound when $r > 1$ for positive advantages), and we skip uninformative prompt groups whose rollouts carry negligible learning signal; **(ii) GSPO-style sequence-level importance sampling** Zheng et al. (2025): we compute the importance ratio at the sequence level and apply clipping at the same granularity as the sequence-level reward, which avoids the mismatch between token-wise importance sampling and sequence-level rewards and stabilizes optimization.

**Objective.** Let $(q, a)$ denote the dialog context and the ground-truth targets, and let $\{o_i\}_{i=1}^{G}$ be $G$ candidates sampled from the behavior policy $\pi_{\theta_{\text{old}}}(\cdot \mid q)$. Each completion $o_i$ is scored by the reward in Sec. §3.2, yielding $R_i \in [0, 1]$. We maximize a *sequence-level, asymmetrically clipped* objective and minimize its negative as the loss:

$$\mathcal{J}_{\text{RL}}(\theta) = \mathbb{E}_{(q,a)\sim\mathcal{D}, \{o_i\}\sim\pi_{\theta_{\text{old}}}(\cdot\mid q)} \left[ \frac{1}{G} \sum_{i=1}^{G} \min\Big(r_i(\theta)\,\hat{A}_i,\ \text{clip}\big(r_i(\theta),\,1-\varepsilon_{\text{low}},\,1+\varepsilon_{\text{high}}\big)\,\hat{A}_i\Big) \right], \tag{26}$$

where $\text{clip}(x, a, b) = \min\{b, \max\{a, x\}\}$ and typically $\varepsilon_{\text{high}} > \varepsilon_{\text{low}}$ ("clip-higher").

**Prompt-group dynamic filtering.** DAPO skips prompt groups whose candidates provide almost no learning signal (e.g., all-correct or all-wrong). Concretely, define batch-normalized advantages and a group-level acceptance criterion:

$$\hat{A}_i = \frac{R_i - \text{mean}(\{R_j\}_{j=1}^{G})}{\text{std}(\{R_j\}_{j=1}^{G})}, \qquad \mathcal{S}(q, a) = \Big\{ i \in \{1, \ldots, G\}\ :\ |\hat{A}_i| > \tau_{\text{adv}} \Big\}, \tag{27}$$

Table 1: Comparison across dimensions (Base, Miss_Func, Miss_Param, Long_Context, Multi-turn Overall) on BFCL v3.

| Models | Method | Base | Miss_Func | Miss_Param | Long_Context | Multi-turn Overall |
|---|---|---|---|---|---|---|
| Llama-3.1-8B-Instruct-FC | Origin | 5.0 | 6.5 | 4.5 | 4.5 | 5.12 |
| | **Ours** | **9.5 (↑95%)** | **7.0 (↑8%)** | **5.0 (↑11%)** | **7.0 (↑56%)** | **7.12 (↑39%)** |
| Qwen2.5-7B-Instruct-FC | Origin | 16.5 | 11.0 | 9.0 | 7.5 | 11.00 |
| | **Ours** | **22.0 (↑33%)** | **13.0 (↑18%)** | **13.5 (↑50%)** | **11.0 (↑47%)** | **14.88 (↑35%)** |
| Qwen3-4B-Instruct | Origin | 18.0 | 19.0 | 13.5 | 14.5 | 16.25 |
| | **Ours** | **25.0 (↑39%)** | **19.5 (↑3%)** | **17.0 (↑26%)** | **21.5 (↑48%)** | **20.75 (↑28%)** |

and require sufficient reward dispersion within the group:

$$\text{Var}\big(\{R_i\}_{i=1}^G\big) > \tau_{\text{var}} \quad \text{and} \quad 0 < |\mathcal{S}(q, a)| < G. \tag{28}$$

If equation 28 fails, we drop the zero-information rollouts and (optionally) draw up to $K$ additional candidates from $\pi_{\theta_{\text{old}}}$, then re-apply the filter. Only indices in $\mathcal{S}(q, a)$ contribute to the expectation in equation 26.

**Sequence-level importance ratio.**  For a completion $o_i = (o_{i,1}, \ldots, o_{i,|o_i|})$, we use the geometric-mean, length-normalized importance ratio:

$$r_i(\theta) = \left( \prod_{t=1}^{|o_i|} \frac{\pi_\theta(o_{i,t} \mid q, o_{i,<t})}{\pi_{\theta_{\text{old}}}(o_{i,t} \mid q, o_{i,<t})} \right)^{1/|o_i|}, \tag{29}$$

and perform clipping at the same sequence granularity as the reward (see equation 26), thereby avoiding token/sequence granularity mismatch.

## 4 EXPERIMENTS

### 4.1 EXPERIMENT SETTINGS

In this part, we will detail the experimental setup, including datasets, hyperparameters, base models, and evaluation metrics.

**Datasets.**  We conduct training on our self-constructed Tool-Reflection-Bench. After human supervision and post-editing, we retained approximately 5k samples in JSONL format to ensure compatibility with RL training under the Swift Zhao et al. (2025b) framework.

**Implementation Details.**  We train models for 1 epoch (a total of 1,000 steps) on 5,000 training samples, using the reward function defined in Sec.3.2. For each training instance, 4 completions were sampled to form a group. The training parameters were set as follows: temperature = 0.85, repetition penalty = 1.1, epsilon = 0.2, epsilon-high = 0.28, with a dynamic sampling strategy adopted.

**Base Models.**  To verify the generalizability of Tool-Reflection-Bench and our training methodology, we conducted experiments using Llama3.1-8B Dubey et al. (2024), Qwen2.5-7B-Instruct Hui et al. (2024), and Qwen3-4B Yang et al. (2025) as base models.

**Evaluation Metrics.**  We evaluated multi-turn tool-calling performance using the Berkeley Function Calling Leaderboard (BFCL) v3 Patil et al., with evaluation dimensions covering multi-turn-base, multi-turn-long-context, multi-turn-miss-func, and multi-turn-miss-param, and the evaluation metric being accuracy. To assess the model's repair capability when tool calls fail, we used Tool-Reflection-Bench, with the evaluation metric being repair rate, Repair@n denotes that for the same data instance, if at least one out of n trials succeeds, the metric is recorded as 1; otherwise, it is 0.

## 4.2 EXPERIMENT RESULTS

### 4.2.1 RESULT ON BFCL V3

**Comparison with base models.**   We conduct performance evaluation on the multi-turn category of BFCL v3 to assess the benefits of enhancing the model's self-reflection capability in multi-turn tool calling, the detailed results are showed in Table. 1. Compared the results against the corresponding base models. The most striking lift appears on **Llama-3.1-8B**: Base rises from 5.0 to 9.5 (+95%) and Long_Context from 4.5 to 7.0 (+56%). **Qwen2.5-7B** shows the largest Miss_Param gain (9.0 $\to$ 13.5, +50%), evidencing stronger parameter repair. **Qwen3-4B** attains an amazing absolute Multi-turn Overall (20.75, +28%) with a sizable Long_Context improvement (+48%). In contrast, its Miss_Func gain is modest ($19.0 \to 19.5$, +3%), indicating tool selection remains comparatively harder—consistent with our method's emphasis on reflection-driven parameter correction and long-context recovery.

### 4.2.2 RESULT ON TOOL-REFLECTION-BENCH

As shown in Table. 2, across open-source baselines, repair rates are low at one try (Repair@1 $\leq$ 9.6%) and only mildly improve with more tries. Training with our method yields consistent gains for all bases: **Llama-3.1-8B-Instruct** jumps from $0.7/5.1/6.8$ to **4.7/20.5/26.4** (Repair@1/3/5), a large improvement especially at higher $n$; **Qwen2.5-7B-Instruct** improves from $2.4/6.1/8.0$ to **9.3/10.3/11.4**; **Qwen3-4B-Instruct** rises from $9.6/10.6/10.6$ to **14.9/18.5/19.5** (best Repair@1 among our models). All finetuned models surpass the closed-source LongCat-Lite-8K-Chat across $n \in \{1, 3, 5\}$, indicating that our reflection-aware reward and RL objective substantially enhance repairability and yield more reliable multi-try recovery. It is also worth noting that when tool calls fail and require repair, our method achieves superior performance compared to **closed-sourced models** of the same scale such as **LongCat-Lite-8K-Chat** Team et al. (2025), **GPT-4o-mini** OpenAI (2024a;b), **GPT-4.1-mini** OpenAI (2025).

Table 2: Experimental Results of Open-Source and Closed-Source Models on the Tool-Reflection-Bench Test Set.

| Models | Repair@1 (%) | Repair@3 (%) | Repair@5 (%) |
|---|---|---|---|
| **Close-Sourced Models** | | | |
| LongCat-Lite-8K-Chat | 2.3 | 3.4 | 4.9 |
| GPT-4o-mini | 6.1 | 8.7 | 9.0 |
| GPT-4.1-mini | 3.1 | 4.3 | 5.1 |
| **Open-Sourced Models** | | | |
| Llama-3.1-8B-Instruct | 0.7 | 5.1 | 6.8 |
| Qwen2.5-7B-Instruct | 2.4 | 6.1 | 8.0 |
| Qwen3-4B-Instruct | 9.6 | 10.6 | 10.6 |
| **Open-Sourced Models Trained on Our Method** | | | |
| **Llama-3.1-8B-Instruct** | **4.7** | **20.5** | **26.4** |
| **Qwen2.5-7B-Instruct** | **9.3** | **10.3** | **11.4** |
| **Qwen3-4B-Instruct** | **14.9** | **18.5** | **19.5** |

## 5 CONCLUSION

This paper proposes a structured reflection method for handling tool call failures, transforming the "from error to repair" process into an explicit, controllable, and trainable action. Our approach overcomes the limitations of previous heuristic, feedback-based self-correction methods in terms of controllability and stability. We further construct Tool-Reflection-Bench for both training and evaluation, and design a task-specific reward function tailored to the tool-calling scenario. In the reinforcement learning stage, we combine the strengths of DAPO and GSPO to enhance training effectiveness. Experimental results show that the proposed method significantly improves multi-turn tool call accuracy on BFCL v3 as well as error repair performance on Tool-Reflection-Bench. Overall, our method and dataset effectively enhance the reliability of tool interactions and offer a new perspective on enabling agents to acquire new capabilities by learning from failure.

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

# A APPENDIX

## A.1 USE OF LLMS

This work leveraged LLMs to verify the mathematical soundness and symbolic accuracy of a few formulas in Sec.A.5.

## A.2 PROMPT FOR PERTURBATION-BASED DISRUPTIONS

In this section, we provide simplified prompts for generating the four types of tool call perturbations, enabling the community to reproduce our setting. The full prompts and implementation code will be released upon the paper's acceptance.

### A.2.1 PROMPT FOR CALL-ORDER SWAP

**How to construct an error tool call example**

**System**

**Goal.** Prepend a controlled erroneous `<call>` and a consistent tool-error message before the first assistant message, so the model must diagnose and repair.

**Procedure.**

1. *Extract calls:* Traverse messages and collect all assistant `<call>...</call>` blocks (regex).

2. *Choose function name:* Parse the *last* call's JSON to get `"name"`; fall back to a regex if needed.

3. *Synthesize wrong call (empty args):*

   ```
   <call>[{"name":"<FUNC_FROM_LAST_CALL>","arguments":{}}]</call>
   ```

4. *Fabricate tool error (pretty JSON string):*

   ```
   {"tool":"<FUNC_FROM_LAST_CALL>","status":"warning",
     "message":"The called function executed but returned partial/mismatched
         ↪ data because the arguments did not match the expected schema for
         ↪ this call.",
     "result": null}
   ```

5. *Insert pair:* Place the wrong assistant call and the tool error *before* the original first assistant message.

6. *Elicit reflection:* Query the LLM with the System/User prompts above to obtain the reflection text, then prepend `<reflect>...</reflect>` to the original assistant message (the original correct call remains).

**Notes.** Using the last call's function ensures schema plausibility; empty arguments induce a controlled failure; the synthetic tool message supplies concrete evidence for the subsequent reflection and repair.

**How to generate a reflection**

**System**

You are an AI assistant that analyzes failed tool calls and provides reflective summaries. Given an original tool call and a fabricated error response, generate a brief reflection explaining why the call likely failed and how to correct it. Be concrete and concise.

**User**

Fill the placeholders {{...}} exactly.
**Original tool call:**

```
{{ORIGINAL_CALL}}
```

**Error response:**

```
{{FAKE_RESPONSE}}
```

Please provide a short reflection on the failure cause and the corrective action.

**An Example**

> **User**
>
> **Original tool call:**
>
> ```
> <call>[{"name":"searchArtistsByArtStyle","arguments":{}}]</call>
> ```
>
> **Error response:**
>
> ```
> {"tool":"searchArtistsByArtStyle","status":"warning",
>  "message":"The called function executed but returned partial/mismatched data
>      ↪ because the arguments did not match the expected schema for this call.",
>  "result": null}
> ```
>
> Please provide a brief reflection on why this tool call failed and what could be improved. Keep it concise and helpful.

### A.2.2 PROMPT FOR REDUNDANT CALL

**How to construct a redundant tool call example**

> **System**
>
> **Goal.** Inject a *redundant* tool call inside an existing `<call>` list and a matching redundant tool response, so the agent must identify and remove the duplication.
> **Procedure.**
>
> 1. *Extract calls:* Traverse the dialogue and collect all assistant-side `<call>...</call>` blocks (regex).
> 2. *Pick a target (not the first):* Uniformly sample an assistant call position from $\{2, \ldots, |\mathcal{C}|\}$.
> 3. *Duplicate within the list:* Parse the target call's JSON. If it is a list, append a deep-copied first element; if it is a single dict, make a two-element list by duplicating it.
> 4. *Fabricate a redundant tool response:* Parse the following `tool` message. Duplicate its first item (or the dict itself) and mark it as redundant, e.g.
>
>    ```
>    {"status":"redundant","message":"This item duplicates a previous result."}
>    ```
>
> 5. *Keep the ground-truth call:* The *correct* call is the original (non-duplicated) first element of the target call list.
> 6. *Place the repair evidence:* After the redundant `tool` message, insert an assistant message with `<reflect>` diagnosing the redundancy and a correct `<call>` (the non-duplicated one), followed by a *clean* tool response (the original, without the redundant copy).
>
> **Notes.** This perturbation preserves schema but injects duplication at both call and response sides, creating a realistic "over-call" pattern for reflection-and-repair.

**How to generate a reflection**

**System**

You are an AI assistant that analyzes *redundant* tool calls and provides reflective summaries. Given a tool-call list and its redundant tool response, write a brief reflection that (i) identifies the duplication, and (ii) states the correct next action (use only the necessary call with proper arguments). Keep the reflection concise and actionable.

**User**

Fill the placeholders {{...}} exactly.
**Tool call list (after duplication):**

{{TOOL_CALL_LIST}}

**Redundant tool response:**

{{REDUNDANT_RESPONSE}}

Please provide a short reflection that points out the redundancy and explains how to proceed correctly.

**An Example**

**User**

**Tool call list (after duplication):**

```
<call>[
  {"name":"searchArtistsByArtStyle","arguments":{"style":"impressionism"}},
  {"name":"searchArtistsByArtStyle","arguments":{"style":"impressionism"}}
]</call>
```

**Redundant tool response:**

```
[
  {"tool":"searchArtistsByArtStyle","status":"ok","items":[...]},
  {"tool":"searchArtistsByArtStyle","status":"redundant",
    "message":"This item duplicates a previous result.","items":[...]}
]
```

Please provide a brief reflection on why this redundant call occurred and how to proceed. Keep it concise and helpful.

### A.2.3 PROMPT FOR MISSING CALL

**How to construct a missing-call perturbation example**

**System**

**Goal.** Remove a necessary assistant `<call>` and make the subsequent call fail due to missing context, so the agent must *recover the omitted call* and then proceed correctly.
**Procedure.**

1. *Extract calls:* Parse all assistant-side `<call>...</call>` blocks (regex).

2. *Select a removable call (not the last):* Uniformly sample an index $i \in \{1, \ldots, |\mathcal{C}|-1\}$.

3. *Find paired tool messages:* Locate the tool reply immediately after call $i$ (the one to remove), and the tool reply after call $i+1$ (the "next" call).

4. *Delete call $i$ and its tool reply.*

5. *Degrade the next call:* For the assistant `<call>` at (original) $i+1$, keep the function but set `"arguments":{}` (empty).

6. *Return an error for the next tool:* Replace that tool reply with an error JSON indicating "missing required arguments".

7. *Reflection and repair insertion:* After the error tool reply, insert:

    (a) an assistant message containing `<reflect>` that explains the omission and a *reinstated* correct `<call>` (the removed call $i$);

    (b) the original tool reply for the removed call $i$;

    (c) the corrected next assistant call (its original, non-empty arguments);

    (d) the corrected next tool reply (its original content).

**Notes.** This perturbation creates a realistic "missing prerequisite call" failure: the subsequent step cannot execute without information from the omitted call. The reflection must (i) identify the omission and (ii) restore the correct call before proceeding.

**How to generate a reflection**

**System**

You are an AI assistant that analyzes *missing* tool calls and provides reflective summaries. Given the omitted call (that should have been executed) and the resulting error response from the next step, write a concise reflection that (i) identifies what was missing, and (ii) states how to proceed: first reinstate the omitted call with correct arguments, then continue.

**User**

Fill the placeholders $\{\{\dots\}\}$ exactly.
**Missing tool call (the one that should have been made):**

`{{MISSING_CALL}}`

**Error response (from the next step):**

`{{ERROR_RESPONSE}}`

Please provide a short reflection that explains the omission and the corrective sequence of actions.

**An Example**

**User**

**Missing tool call:**

`<call>[{"name":"fetchUserProfile","arguments":{"user_id":"u_1293"}}]</call>`

**Error response (from the next step):**

```
[
  {"status":"error",
   "message":"Missing required arguments. The function call failed because necessary
   ↪    parameters were not provided.",
   "result": null}
]
```

Please provide a brief reflection on what was missing and how to proceed. Keep it concise and helpful.

A.2.4   PROMPT FOR ARGUMENT ERROR

**How to construct an argument–error perturbation example**

**System**

**Goal.** Corrupt the arguments of an existing assistant `<call>` so that the paired tool reply returns a parameter–validation error, forcing the agent to *diagnose mismatched/invalid arguments* and repair with the correct call.

**Procedure.**

1. *Extract calls:* Parse all assistant-side `<call>...</call>` blocks via regex.

2. *Select a call:* Uniformly sample one index $i \in \{1, \ldots, |\mathcal{C}|\}$ and locate its immediate tool reply.

3. *Corrupt arguments:* Keep `"name"` unchanged; replace `"arguments"` with perturbed values (e.g., wrong types, out-of-range numbers, empty strings, unknown keys). The JSON stays well-formed:

   `<call>[{"name":"<FUNC_NAME>","arguments":{<WRONG_ARGS>}}]</call>`

4. *Synthesize error reply:* Replace the paired tool message with a structured error indicating invalid parameters (e.g., `"error_code":"INVALID_PARAMETERS"` and an informative message).

5. *Reflection and repair insertion:* Immediately after the error, insert:

   (a) an assistant message with `<reflect>` that contrasts the wrong vs. correct arguments and states the fix;

   (b) the *original* (correct) call and its original (successful) tool reply.

**Notes.** Do not alter the function name; only arguments are corrupted. Keep JSON/tags valid to isolate the failure mode to argument errors.

**How to generate a reflection**

**System**

You are an AI assistant that analyzes incorrect tool-call *parameters* and provides a reflective summary. Given the correct call, the wrong call (with corrupted arguments), and the error response, write a brief reflection that (i) pinpoints which arguments are incorrect and why, and (ii) states the corrected call. Be concrete and concise.

**User**

Fill the placeholders {{...}} exactly.
**Correct tool call (ground truth):**

`{{CORRECT_CALL}}`

**Wrong tool call made (arguments corrupted):**

`{{WRONG_CALL}}`

**Error response:**

`{{ERROR_RESPONSE}}`

Please provide a short reflection that identifies the parameter issues and the corrective action.

**An Example**

**User**

**Correct tool call:**

```
<call>[{"name":"bookFlight",
      "arguments":{"from":"SFO","to":"JFK","date":"2025-11-02","passengers
   ↪ ":1}}]</call>
```



**Wrong tool call made:**

```
<call>[{"name":"bookFlight",
        "arguments":{"from":999999,"to":"","date":null,"passengers":"many"}}]</call
    ↪ >
```

**Error response:**

```
[{"status":"error",
  "message":"Parameter validation failed for bookFlight. One or more arguments are
      ↪ invalid.",
  "result": null,
  "error_code":"INVALID_PARAMETERS"}]
```

Please provide a brief reflection on which parameters are incorrect and how to fix them. Keep it concise and helpful.



## A.3    TRAINING DATA CASE STUDY

In this section, we extract one complete sample from each of the four perturbation modes for analysis. Due to space limitations, these four samples are provided in the supplementary material for reference, while here we only present a brief analysis of the data.

### A.3.1    CASE STUDY OF CALL-ORDER SWAP

**Setup.**   The user requests end–to–end logistics for a 10–person business trip (NYC $\rightarrow$ MIA): search and *book* round–trip flights, search and *book* hotel rooms, and arrange airport–hotel ground transportation. The toolset exposes search_flights, book_flight, search_hotels, book_hotel, and arrange_transportation, where the last function *depends* on confirmed itinerary details (arrival time and hotel).

**Failure trace (order dependency violated).**   The agent correctly issues two discovery calls, then prematurely attempts to arrange transportation *before* booking:

```
<tool_call>{"name":"search_flights", ...}</tool_call>
<tool_call>{"name":"search_hotels",  ...}</tool_call>

<tool_call>{"name":"arrange_transportation",
            "arguments":{"arrival_date":"2023-03-21",
                         "passenger_count":10,
                         "pickup_location":"MIA Airport",
                         "dropoff_location":"(to be decided)"}}</tool_call>
```

The tool returns a dependency error because dropoff_location cannot be finalized prior to book_hotel and pickup timing should be aligned to book_flight:

```
Tool: {"status":"error",
       "message":"Transportation requires confirmed flight and hotel details.",
       "result": null}
```

**Model reflection and repair.**   The agent emits a concise <reflect> that identifies the *order–dependency* (transport must follow booking), then repairs by booking first and only then arranging transport:

```
<reflect>
Failure stems from an order dependency: transportation must follow confirmed
flight and hotel, as pickup/dropoff depend on the booked itinerary.
Correct plan: (1) book flight; (2) book hotel; (3) arrange transportation.
</reflect>

<tool_call>{"name":"book_flight", ...}</tool_call>
<tool_call>{"name":"book_hotel",  ...}</tool_call>
```

```
<tool_call>{"name":"arrange_transportation",
            "arguments":{"arrival_date":"2023-03-21",
                        "passenger_count":10,
                        "pickup_location":"MIA",
                        "dropoff_location":"Oceanview Hotel"}}</tool_call>
```

**Why the reward favors the fix.** Under our reward in Sec. §3.2: (i) the reflection receives a high semantic score $s_{\text{ref}}$ for explicitly diagnosing the *order* constraint; (ii) the repaired calls exactly match schema and parameters, yielding $s_{\text{call}} = 1$ (order–*invariant* matching across the multiset of calls), while the *format/penalty factor* $F$ upweights the repaired trajectory because executability and dependency checks now pass; (iii) the premature arrange_transportation in the failed rollout triggers penalties in $F$ (dependency/executability), so $R_{\text{core}}$ is suppressed and the dynamic sampling filter (Sec. §3.3) down–weights such uninformative rollouts. Overall, the corrected plan yields a strictly larger $R_{\text{total}}$ than the erroneous one.

**Summary.** This example shows that explicit reflection converts a latent, long–range constraint ("book before transport") into a short textual diagnosis that the policy can optimize against. Our order–invariant call equality avoids over–penalizing benign output permutations, while dependency–aware format shaping in $F$ enforces *inter–call* preconditions. Empirically, such cases account for a substantial share of multi–turn gains, whereas single–turn metrics remain stable.

### A.3.2 CASE STUDY OF REDUNDANT CALL

**Setup and failure mode.** The user requests the list of services for all NYC banks operating for $> 10$ years. The agent first queries the bank roster with get_operational_banks, then correctly queries services for B001, but erroneously *repeats* the same service lookup for B001 instead of proceeding to unseen banks (B002--B004). This is a canonical *redundant* call error: wasted tool budget without adding new information.

```
...
<tool_call>
{"name":"get_operational_banks",
 "arguments":{"city":"New York City","min_years_operational":10}}
</tool_call>

User (tool): [{"name":"get_operational_banks","results":{"banks":[
  {"bank_id":"B001"}, {"bank_id":"B002"}, {"bank_id":"B003"}, {"bank_id":"B004"}]}}]

<tool_call>
{"name":"get_bank_services","arguments":{"bank_id":"B001"}}
</tool_call>

User (tool): [{"name":"get_bank_services","arguments":{"bank_id":"B001"},
 "results":{"services":[...]} }]

# Redundant (erroneous) repeat:
<tool_call>
{"name":"get_bank_services","arguments":{"bank_id":"B001"}}
</tool_call>

User (tool): [{"name":"get_bank_services","arguments":{"bank_id":"B001"},
 "results":{"services":[...], "note":"Duplicate request"}}]
...
```

**Reflection and repair (ground truth).** The reflection diagnoses duplication and prescribes de-duplication; the corrected plan proceeds with the remaining, unseen IDs:

```
<reflect>
The last call repeated B001, yielding redundant information.
```

```
1026    Avoid duplicates; continue with B002, B003, B004 exactly once each.
1027    </reflect>
1028
1029    <tool_call>{"name":"get_bank_services","arguments":{"bank_id":"B002"}}</tool_call>
1030    <tool_call>{"name":"get_bank_services","arguments":{"bank_id":"B003"}}</tool_call>
1031    <tool_call>{"name":"get_bank_services","arguments":{"bank_id":"B004"}}</tool_call>
```

**Why the model failed.** The failure arises from (i) insufficient state tracking over the set of already-seen entities (here, bank IDs), and (ii) weak inductive bias against issuing calls whose *marginal information gain* is near zero. In multi-turn settings, local myopic policies often re-issue the last successful pattern without cross-step deduplication.

**How the reward steers recovery.** Our scoring treats call sets as order-invariant but schema-strict; redundant calls trigger the count-mismatch component in the format factor $F$ (penalizing $|C_{\text{calls}}| \neq |G_{\text{calls}}|$) while EqualCalls fails due to multiset mismatch. The reflection text receives a positive semantic score if it explicitly identifies the duplication and prescribes the missing IDs, encouraging concise, actionable self-correction. Together, the structure score $S$ and format factor $F$ downweight redundant completions and upweight the repaired sequence.

**Summary.** This case shows that explicit reflection converts a silent efficiency bug into a supervised correction step: the agent (1) cites the duplicated identifier, (2) enumerates the remaining targets, and (3) completes them exactly once. Empirically, such reflection-shaped supervision reduces redundant tool usage and improves multi-turn success without harming single-turn accuracy.

### A.3.3 Case Study of Missing Call

**Setup.** The user asks to register *four* tax documents: (i) W-2 (ABC Corp), (ii) 1099-INT (First National Bank), (iii) property tax statement (county assessor), and (iv) Form 1098 (mortgage lender). The tool schema exposes a single function add_tax_documents(name, value, category, priority) with name, value required.

**Baseline failure (*missing calls*).** The baseline assistant emits only two <tool_call>s (W-2, 1099-INT) and then stops, yielding a 50% recall on required calls. Formally, let $G_{\text{calls}}$ contain the four intended calls and $C_{\text{calls}}$ the two produced calls. Then $|G_{\text{calls}}| = 4$, $|C_{\text{calls}}| = 2$, and the call–set equality test fails: $\text{EqualCalls}(C_{\text{calls}}, G_{\text{calls}}) = 0$. This is a typical *missing-call* error in multi-item requests: the model recognizes the pattern "one item $\rightarrow$ one call" but truncates the sequence, leaving later items unprocessed.

**Structured reflection (*diagnosis*).** Our method takes the partially executed trajectory as *negative evidence* and the original request as *positive intent* and generates an explicit reflection:

> <reflect> "I missed 2 tool call(s). The user listed multiple items, and each item requires a separate call. I should enumerate all items and complete the remaining calls." </reflect>

The reflection correctly localizes the failure (under-counting of required calls), quantifies the deficit (missed= 2), and states the repair rule (enumerate all items $\Rightarrow$ one call per item).

**Repairs (*corrective calls*).** Conditioned on the reflection, the agent appends the missing tool calls for the remaining items:

- name: Property tax statement; value: county assessor record; category: personal;
- name: Form 1098; value: mortgage interest statement; category: personal.

The assignments work$\rightarrow$W-2,1099-INT and personal$\rightarrow$property tax, 1098 are semantically consistent: the former are employment/bank income records; the latter are household liabilities/taxes.*(Any schema-compatible categorization would pass executability; ours also preserves natural semantics.)*

**Why this matters.** This case highlights a frequent multi-turn brittleness: once the agent produces a plausible prefix of calls, it prematurely concludes and fails to cover all requested items. By making *missingness* an explicit, trainable concept, structured reflection converts a sparse binary signal (success/failure) into actionable supervision:

1. **Detection:** Compare item cardinalities and arguments; compute $\mathbb{I}[|C_{\text{calls}}| < |G_{\text{calls}}|]$ and list uncovered entities.

2. **Diagnosis:** Attribute the error to *enumeration/coverage* rather than formatting or parameters.

3. **Repair:** Synthesize the exact missing calls with schema-valid arguments; preserve already-correct calls.

**Summary.** Empirically, such instances improve the model's *coverage discipline*: after training, we observe higher multi-item completion rates with negligible increase in redundant calls, indicating that the model learned "one-mentioned-item $\Rightarrow$ one-call" as a robust policy rather than overcalling.

### A.3.4 CASE STUDY OF ARGUMENT ERROR

**Setup.** The tool schema exposes multiple functions with *schema–strict* parameters (e.g., `check_plant_water_level(plant_location:string)`, `start_watering(plant_location:string, duration:number)`, `start_trimming(hedge_location:string)`, ...). The user requests two primary actions in the backyard: (i) trim hedges and (ii) water all potted plants for about 10 minutes; afterwards ensure plants have enough water and dispose clippings.

**Baseline failure (*argument error*).** The assistant issues

```
<call>[{"name":"check_plant_water_level","arguments":{}}]</call>
```

omitting the *required* key `plant_location`. The tool returns a schema warning that the arguments "did not match expected schema." Under our reward, the call-level indicator $s_{\text{call}}$ is 0 because the produced call fails schema equality (tool name matches, but the argument map does not).

**Structured reflection (*diagnosis*).** The reflection generated by our process states that the call "failed because it did not include the required arguments needed by the function's schema," and prescribes: "ensure all necessary parameters are provided according to the function's documentation." This localizes the error to **parameter mis-specification** (not tool selection or ordering), and points to the concrete fix—satisfy the schema.

**Repairs (*efficient plan consistent with the request*).** Given the user's 10-minute target and the backyard scope, the corrected action set executes the two core operations with schema-valid arguments:

- `start_watering(plant_location="backyard", duration=10)`
- `start_trimming(hedge_location="backyard")`

These can be dispatched in parallel (independent resources), achieving the requested time budget while ensuring plants receive sufficient water and hedges are trimmed. This replaces the invalid pre-check with a direct, time-bounded watering call that already satisfies the user's constraint.

**Why this matters.** Argument errors are common in tool use and typically yield *sparse* feedback ("schema mismatch"). By forcing the model to (i) recognize the missing required field and (ii) restate the schema-conformant fix, the reflection step converts a low-information error into actionable supervision. In our benchmark, such instances consistently improve:

1. **Schema adherence:** higher exact-match rate on `name`/`arguments`.

2. **Planning under constraints:** selection of parameterized calls (`duration=10`) aligned with user constraints instead of brittle pre-checks with empty arguments.

3. **Stability:** fewer retries and warnings downstream because calls are executable on the first attempt.

**Summary.** This case illustrates how reflection-guided repair turns a malformed `<call>` into a compact, correct, and time-efficient action plan.

### A.4 TEST DATA CASE STUDY

In this section, we present two representative test cases and their corresponding evaluation results as a case study, providing an intuitive demonstration of the effectiveness of our method and the model's self-reflection capability for tool-call repair. Since the original cases are relatively long, we include their full content in the supplementary material for reference and provide only the analysis here.

#### A.4.1 CASE I

**Setting.** The tool set exposes three functions: `getRecipes(max_time, meal_type)`, `getSmoothieIngredients(max_time)`, and `findComplementaryRecipes(recipes, ingredients)`. The user asks for *breakfast* recipes under 15 minutes and smoothie pairings under 5 minutes.

**Failure mode (pre-training).** The baseline model immediately issues

```
[{"name":"findComplementaryRecipes","parameters":{}}]
```

which violates the function schema (both `recipes` and `ingredients` are required). The tool returns a schema-warning. Under our reward, this yields $s_{\text{call}} = 0$ and triggers format penalties $F < 1$ due to missing required parameters.

**Reflection-driven repair (post-training).** After RL on Tool-Reflection-Bench, the model first *reflects* that the failure arises from absent inputs, then correctly decomposes the task into *produce inputs → compose*:

```
[{"name":"getRecipes","parameters":{"max_time":15,"meal_type":"breakfast"}}]
[{"name":"getSmoothieIngredients","parameters":{"max_time":5}}]
[{"name":"findComplementaryRecipes",
  "parameters":{"recipes": <from getRecipes>, "ingredients": <from
    ↪ getSmoothieIngredients>}}]
```

This satisfies the schema strictly (tool names and parameter maps match), making the call set correct and executable.

**Why our method helps.** (i) **Reward shaping:** The instance accrues a hard penalty when required fields are absent; after repair, $s_{\text{call}}$ flips to $1$ and $F \to 1$, raising $R_{\text{core}} = S \cdot F$ substantially. (ii) **Sequence-level RL:** The GSPO-style sequence-ratio with dual clipping aligns the optimization granularity with the sequence reward, while DAPO-style dynamic filtering removes near-zero-advantage rollouts (all-wrong/all-correct), sharpening learning signals for this failure mode.

**Takeaway.** Compared to the baseline that *jumps* to composition with empty inputs, the trained policy learns to (a) diagnose the schema error, (b) *stage* prerequisite calls to produce the missing inputs, and (c) complete the composition with a schema-valid call set. This precisely matches our benchmark's objective: enable robust, multi-turn tool use via reflection and repair.

#### A.4.2 CASE II

**Setting.** Available tools include `get_current_season()`, `get_seeds_by_season(season)`, `filter_seeds_by_availability(seeds)`, `purchase_seeds(seeds, quantity)`, and `calculate_total_cost(purchased_seeds)`. The user asks to *buy 10 packets of seasonal vegetable seeds* and *report the total cost*.

**Failure mode (pre-training).** The baseline calls the aggregator first, with no inputs:

```
[{"name":"calculate_total_cost","parameters":{}}]
```

This violates the required schema (purchased_seeds missing), producing a warning and yielding $s_{\text{call}} = 0$ and a strong format penalty $F < 1$ in our reward.

**Reflection-driven repair (post-training).** After RL on Tool-Reflection-Bench, the model first *reflects* that costing requires purchased items, then executes a staged pipeline to materialize prerequisites before aggregation:

```
[{"name":"get_current_season","parameters":{}}]
```

```
[{"name":"get_seeds_by_season","parameters":{"season":"<CUR_SEASON>"}}]
```

```
[{"name":"filter_seeds_by_availability","parameters":{"seeds":<SEASONAL_SEEDS>}}]
```

```
[{"name":"purchase_seeds","parameters":{"seeds":<AVAILABLE_SEEDS>,"quantity":10}}]
```

```
[{"name":"calculate_total_cost","parameters":{"purchased_seeds":<PURCHASED>}}]
```

Each call now matches tool name and parameter map exactly (schema-strict), so $s_{\text{call}} = 1$ and $F \to 1$.

**Why it works.** *Reward design* penalizes missing required fields and redundant structure, while granting full credit only when the `<call>` set exactly matches the ground truth (schema-strict, order-invariant). The *sequence-level RL objective* (GSPO-style ratio, dual clipping) aligns optimization with sequence rewards, and *DAPO-style dynamic filtering* removes near-zero-advantage groups, concentrating updates on informative failures. Together these guide the policy to diagnose schema errors, stage prerequisite calls, and complete the costing correctly.

**Takeaway.** The trained policy no longer "guesses" totals from empty inputs. Instead, it *plans → acquires data → purchases → aggregates*, a behavior precisely targeted by our reflection-and-repair rewards.

## A.5 THEORETICAL ANALYSIS

We analyze the main design choices of our reward in Sec. §3.2 and the RL objective in Sec. §3.3. Throughout, $\text{Sim} \in [0, 1]$, all weights are nonnegative, presence masks are indicators, and $\text{clip}(x, a, b) = \min\{b, \max\{a, x\}\}$. *To avoid symbol overloading, we denote by $r_{\text{fmt}}$ the format-penalty attenuation scalar used in Sec. §3.2 (called $r$ there), and by $r_{\text{seq}}$ the sequence-level importance ratio in Sec. §3.3.*

### A.5.1 CONSISTENCY OF PRESENCE-MASK NORMALIZATION

Recall

$$S = \frac{w_{\text{r}} I_{\text{r}} \, s_{\text{ref}} + w_{\text{c}} I_{\text{c}} \, s_{\text{call}} + w_{\text{f}} I_{\text{f}} \, s_{\text{final}}}{W_{\text{act}}}, \qquad W_{\text{act}} = w_{\text{r}} I_{\text{r}} + w_{\text{c}} I_{\text{c}} + w_{\text{f}} I_{\text{f}}, \tag{30}$$

where $w_{\bullet} \geq 0$, $I_{\bullet} \in \{0, 1\}$, at least one $I_{\bullet} = 1$, $s_{\text{ref}}, s_{\text{final}} \in [0, 1]$, and $s_{\text{call}} \in \{0, 1\}$.

**Lemma 1 (Convex-combination form).** Let $\mathcal{A} = \{k \in \{\text{r,c,f}\} : I_k = 1\}$ and define

$$\alpha_k = \frac{w_k}{\sum_{j \in \mathcal{A}} w_j} \quad \text{for } k \in \mathcal{A}. \tag{31}$$

Then $\alpha_k \geq 0$, $\sum_{k \in \mathcal{A}} \alpha_k = 1$, and

$$S = \sum_{k \in \mathcal{A}} \alpha_k \, s_k \quad \text{with } s_{\text{r}} = s_{\text{ref}}, \ s_{\text{c}} = s_{\text{call}}, \ s_{\text{f}} = s_{\text{final}}. \tag{32}$$

*Proof.* Since $I_k = 1$ iff $k \in \mathcal{A}$, the numerator equals $\sum_{k \in \mathcal{A}} w_k s_k$ and $W_{\text{act}} = \sum_{k \in \mathcal{A}} w_k > 0$. Divide both to obtain the stated form.

**Proposition 1 (Boundedness, stability, and scale invariance).** With $W_{\text{act}} > 0$:

    *(a)* $S \in [0, 1]$ and, more sharply, $S \in [\min_{k \in \mathcal{A}} s_k, \ \max_{k \in \mathcal{A}} s_k]$.

    *(b)* If one only toggles *absent* parts (keeps $\mathcal{A}$ and $\{w_k\}_{k \in \mathcal{A}}$ unchanged), then $S$ is unchanged.

    *(c)* For any $\lambda > 0$, replacing each active weight by $\lambda w_k$ leaves $S$ unchanged.

*Proof.* (a) By Lemma 1, $S$ is a convex combination of $\{s_k\}_{k \in \mathcal{A}}$; the interval bound follows from $s_k \in [0, 1]$. (b) Absent-part toggles do not change $\mathcal{A}$ nor the active $w_k$. (c) Common scaling cancels in numerator/denominator.

**Corollary 1 (Continuity and Lipschitzness).** Fix $\mathcal{A}$ and $w_k$ for $k \in \mathcal{A}$. Then $S$ is an affine (hence continuous) map of $(s_k)_{k \in \mathcal{A}}$ with

$$|S - S'| \ \leq \ \sum_{k \in \mathcal{A}} \alpha_k \, |s_k - s'_k| \ \leq \ \max_{k \in \mathcal{A}} |s_k - s'_k|, \tag{33}$$

so $S$ is 1-Lipschitz w.r.t. the $\ell_\infty$-norm on the active scores.

**Remark.** The definition via $\text{clip}_{[0,1]}(\cdot)$ in equation 25 is not needed for $S$ since the convex-combination form already implies $S \in [0, 1]$.

### A.5.2    FORMAT FACTOR: BOUNDEDNESS, MONOTONICITY, AND EQUALCALLS ATTENUATION

Let

$$P_{\text{total}} \ = \ P_{\text{miss}} \ + \ \beta_{\text{extra}} \, P_{\text{extra}} \ + \ \gamma_{\text{count}} \, P_{\text{count}}, \qquad \beta_{\text{extra}}, \gamma_{\text{count}} \geq 0, \quad P_\bullet \geq 0, \tag{34}$$

and define the attenuation scalar

$$r_{\text{fmt}} \ = \ \begin{cases} r_{\text{reduce}}, & \text{EqualCalls}(C_{\text{calls}}, G_{\text{calls}}), \\ 1, & \text{otherwise}, \end{cases} \qquad r_{\text{reduce}} \in (0, 1]. \tag{35}$$

Consider

$$F \ = \ \text{clip}_{[0,1]}(1 - \lambda_m \, P_{\text{total}} \, r_{\text{fmt}}), \qquad \lambda_m \geq 0. \tag{36}$$

This is equivalent to the piecewise definition in equation 23 since $P_{\text{miss}} = P_{\text{extra}} = P_{\text{count}} = 0$ implies the inner value equals 1.

**Proposition 2 (Core properties of $F$).**

    *(a) Boundedness and regularity.* $F \in [0, 1]$ for all inputs; $F$ is continuous, piecewise affine in $(P_{\text{miss}}, P_{\text{extra}}, P_{\text{count}})$ and 1-Lipschitz w.r.t. its scalar argument before clipping.

    *(b) Monotonicity.* For fixed $(\lambda_m, r_{\text{fmt}})$, $F$ is nonincreasing in $P_{\text{miss}}, P_{\text{extra}}, P_{\text{count}}$ and nonincreasing in $\lambda_m$ and in $r_{\text{fmt}}$.

    *(c) EqualCalls attenuation improves $F$.* If EqualCalls holds so that $r_{\text{fmt}}$ is replaced by $r_{\text{reduce}} \leq 1$, then $F$ weakly increases.

    *(d) Plateau characterization.* $F = 1$ iff $\lambda_m P_{\text{total}} r_{\text{fmt}} = 0$ (e.g., $P_{\text{total}} = 0$ or $\lambda_m = 0$). If $\lambda_m > 0$ and $r_{\text{fmt}} > 0$, then $F = 0$ iff $P_{\text{total}} \geq 1/(\lambda_m r_{\text{fmt}})$.

**Corollary 2 (Sensitivity bound).** Off the plateaus ($1 - \lambda_m P_{\text{total}} r_{\text{fmt}} \in (0, 1)$),

$$|\Delta F| \ \leq \ \lambda_m r_{\text{fmt}} \left( |\Delta P_{\text{miss}}| + \beta_{\text{extra}} |\Delta P_{\text{extra}}| + \gamma_{\text{count}} |\Delta P_{\text{count}}| \right). \tag{37}$$

### A.5.3    CORE REWARD WITH SIMILARITY BACKOFF: SIGNAL AND VARIANCE CONTROL

Let $R_{\text{core}} = S \cdot F$ as in equation 24. The total reward uses a backoff when $R_{\text{core}}$ is very small:

$$R_{\text{total}} \ = \ \begin{cases} \text{clip}_{[0,1]}(R_{\text{core}}), & R_{\text{core}} \geq \varepsilon, \\ \text{clip}_{[0,1]}\left(w_{\text{b}} \cdot \text{Sim}(\text{concat}(C), \text{concat}(G))\right), & \text{otherwise}, \end{cases} \tag{38}$$

with $w_{\text{b}} \in (0, 1]$ and $\varepsilon > 0$. Note $R_{\text{core}} \in [0, 1]$ already, hence clipping is redundant but harmless and keeps the two branches notationally symmetric.

We analyze its effect under a standard policy-gradient estimator $\nabla_\theta \mathbb{E}[R_{\text{total}}] = \mathbb{E}[R_{\text{total}} \, \nabla_\theta \log \pi_\theta(\cdot)]$.

**Lemma 3 (Uniform bounded variance of the reward).** Since $R_{\text{total}} \in [0,1]$, we have $\text{Var}(R_{\text{total}}) \leq \frac{1}{4}$ for any data distribution.

**Lemma 4 (Non-degenerate gradient second moment on the backoff branch).** Let $\mathcal{B} = \{R_{\text{core}} < \varepsilon\}$ with $\mathbb{P}(\mathcal{B}) = p > 0$. Assume $\text{Sim}(\text{concat}(C), \text{concat}(G)) \geq \sigma$ a.s. on $\mathcal{B}$ for some $\sigma > 0$, and $\mathbb{E}\big[\|\nabla_\theta \log \pi_\theta(\cdot)\|^2 \, \mathbf{1}_{\mathcal{B}}\big] > 0$. Then

$$\mathbb{E}\Big[\, \big\|R_{\text{total}} \, \nabla_\theta \log \pi_\theta(\cdot)\big\|^2 \,\Big] \;\geq\; (w_{\text{b}}\sigma)^2 \, \mathbb{E}\Big[\, \big\|\nabla_\theta \log \pi_\theta(\cdot)\big\|^2 \, \mathbf{1}_{\mathcal{B}} \,\Big] \;>\; 0. \tag{39}$$

*Implication.* When $R_{\text{core}}$ requently approaches 0 (in the early stages of training), the backoff branch ensures that the second moment of the gradients does not degenerate; combined with the variance upper bound from Lemma 3, this helps stabilize the optimization updates.

### A.5.4 Sequence-Level Importance Sampling and Clipping

Let the sampled completion be $o = (o_1, \ldots, o_T)$, and define the sequence-level (geometric-mean, length-normalized) ratio

$$r_{\text{seq}}(\theta) = \left(\prod_{t=1}^T \frac{\pi_\theta(o_t \mid q, \, o_{<t})}{\pi_{\theta_{\text{old}}}(o_t \mid q, \, o_{<t})}\right)^{1/T} \;=\; \exp\!\Big(\tfrac{1}{T}\sum_{t=1}^T \log \rho_t\Big), \quad \rho_t = \frac{\pi_\theta(o_t \mid \cdot)}{\pi_{\theta_{\text{old}}}(o_t \mid \cdot)}. \tag{40}$$

**Proposition 3 (Length-independent ratio range under bounded log-ratios).** If $\log \rho_t \in [-L, L]$ a.s. for some $L > 0$, then

$$e^{-L} \;\leq\; r_{\text{seq}}(\theta) \;\leq\; e^L \qquad \text{for all } T \geq 1, \tag{41}$$

whereas the unnormalized product ratio ranges in $[e^{-LT}, e^{LT}]$.

*Implication.* The geometric mean aligns the *ratio granularity* with the *sequence-level reward* in equation 26, prevents exponential blow-up with $T$, and—together with dual clipping—reduces variance at the sequence level.

### A.5.5 Dynamic Filtering of Prompt Groups (DAPO-style)

Let a prompt group produce $G$ rollouts $\{o_i\}_{i=1}^G$ with rewards $R_i \in [0,1]$ and batch $z$-scored advantages

$$\hat{A}_i \;=\; \frac{R_i - \bar{R}}{s_R}, \qquad \bar{R} = \tfrac{1}{G}\sum_{j=1}^G R_j, \;\; s_R = \sqrt{\tfrac{1}{G}\sum_{j=1}^G (R_j - \bar{R})^2} \;>\; 0. \tag{42}$$

Define the *accepted* set

$$\mathcal{S} \;=\; \big\{\, i : |\hat{A}_i| > \tau_{\text{adv}} \,\big\}, \qquad 0 < |\mathcal{S}| < G, \qquad \text{Var}(\{R_i\}_{i=1}^G) > \tau_{\text{var}} > 0. \tag{43}$$

Write the per-sample (sequence-level, dual-clipped) PPO-like term as

$$\ell_i(\theta) \;=\; \min\Big\{\, r_{\text{seq},i}(\theta)\, \hat{A}_i, \; \text{clip}\big(r_{\text{seq},i}(\theta), 1 - \varepsilon_{\text{low}}, 1 + \varepsilon_{\text{high}}\big)\, \hat{A}_i \,\Big\}, \tag{44}$$

and denote its gradient by $g_i(\theta) = \nabla_\theta \ell_i(\theta)$. Assume the usual score-function bound and clipped ratio range:

$$\big\|\nabla_\theta \log \pi_\theta(o_{i,t} \mid q, o_{i,<t})\big\| \;\leq\; B_\pi, \quad r_{\text{seq},i}(\theta) \in [\, 1 - \varepsilon_{\text{low}}, \, 1 + \varepsilon_{\text{high}}\,]. \tag{$\star$}$$

**A uniform bound on per-rollout gradients.** Since $r_{\text{seq},i}(\theta)$ is the geometric mean of token ratios,

$$\nabla_\theta r_{\text{seq},i}(\theta) \;=\; r_{\text{seq},i}(\theta)\, \frac{1}{|o_i|} \sum_{t=1}^{|o_i|} \nabla_\theta \log \pi_\theta(o_{i,t} \mid q, o_{i,<t}). \tag{45}$$

Using ($\star$) and that the clipped branch is constant on plateaus, there exists a finite $C_\psi = (1 + \varepsilon_{\text{high}}) B_\pi$ such that

$$\|g_i(\theta)\| \;\leq\; C_\psi \, |\hat{A}_i| \qquad \text{for all } i, \, \theta. \tag{46}$$

**Lemma 5 (Zero or near-zero advantages).** (a) If $\hat{A}_i = 0$, removing $o_i$ leaves the group-wise expected gradient unchanged.

(b) If $|\hat{A}_i| \le \tau_{\mathrm{adv}}$, then, for any $\theta$,

$$\big\| \, \mathbb{E}\,[\, g_i(\theta)\,]\, \big\| \;\le\; C_\psi \, \tau_{\mathrm{adv}}, \qquad \mathbb{E}\big[\, \|g_i(\theta)\|^2\,\big] \;\le\; C_\psi^2\, \tau_{\mathrm{adv}}^2. \tag{47}$$

*Proof.* (a) The contribution is proportional to $\hat{A}_i$. (b) Apply equation 46 and take expectations.

**Bias and variance effects with $\frac{1}{G}$ normalization.** Let the *filtered* group gradient be

$$\tilde{g}(\theta) \;=\; \frac{1}{G}\sum_{i\in\mathcal{S}} g_i(\theta), \qquad g(\theta) \;=\; \frac{1}{G}\sum_{i=1}^{G} g_i(\theta)\ \text{(unfiltered)}. \tag{48}$$

Define the discarded set $\mathcal{S}^c = \{1,\dots,G\} \setminus \mathcal{S}$. Then

$$\mathbb{E}[\tilde{g}(\theta)] - \mathbb{E}[g(\theta)] \;=\; -\frac{1}{G}\sum_{i\in\mathcal{S}^c} \mathbb{E}[\, g_i(\theta)\,], \tag{49}$$

$$\big\| \, \mathbb{E}[\tilde{g}(\theta)] - \mathbb{E}[g(\theta)]\, \big\| \;\le\; \frac{|\mathcal{S}^c|}{G}\, C_\psi\, \tau_{\mathrm{adv}} \;\le\; C_\psi\, \tau_{\mathrm{adv}},$$

using Lemma 5(b). Moreover,

$$\mathbb{E}\left[\, \Big\| \frac{1}{G}\sum_{i\in\mathcal{S}^c} g_i(\theta) \Big\|^2 \,\right] \;\le\; \frac{|\mathcal{S}^c|}{G^2}\, C_\psi^2\, \tau_{\mathrm{adv}}^2, \tag{50}$$

thus, discarding near-zero advantageous terms induces at most an $O(\tau_{\mathrm{adv}}^2)$-level change in the second moment; with respect to the $\frac{1}{G}$ normalization, it does not introduce any additional scaling bias.

**Acceptance constraints avoid degeneracy.** The constraints $0 < |\mathcal{S}| < G$ and $\mathrm{Var}(\{R_i\}) > \tau_{\mathrm{var}}$ ensure: (i) the batch standardization $s_R$ is well-defined; (ii) both positive and negative (or at least non-identical) signals are present, preventing the trivial zero-gradient case where all $\hat{A}_i$ are identical. Consequently, $\tilde{g}(\theta)$ is a non-degenerate direction whenever useful learning signal exists.

**Asymptotic unbiasedness with vanishing threshold.** If the threshold decays $\tau_{\mathrm{adv}}^{(t)} \downarrow 0$ and the law of $\hat{A}_i$ has a continuous density at 0, then the discard probability $\mathbb{P}(|\hat{A}_i| \le \tau_{\mathrm{adv}}^{(t)}) \to 0$, and

$$\lim_{t\to\infty} \big\| \, \mathbb{E}[\tilde{g}_t(\theta)] - \mathbb{E}[g(\theta)]\, \big\| \;=\; 0, \tag{51}$$

i.e., the dynamic filtering becomes asymptotically unbiased while retaining finite-time variance-reduction benefits.

**Summary.** Dynamic filtering deletes rollouts whose contributions are provably negligible (zero or $O(\tau_{\mathrm{adv}})$), thereby reducing variance and compute without altering the expected update in the limit $\tau_{\mathrm{adv}} \to 0$; using the same $1/G$ normalization as equation 26 avoids spurious scaling bias.

A.5.6 CONVERGENCE CONSIDERATIONS FOR THE CLIPPED SEQUENCE-LEVEL OBJECTIVE

Consider the surrogate objective $\mathcal{J}_{\mathrm{RL}}(\theta)$ in equation 26, where rewards are bounded in $[0,1]$ and the sequence-level importance ratios are dual-clipped to $[\,1 - \varepsilon_{\mathrm{low}},\, 1 + \varepsilon_{\mathrm{high}}\,]$.

**Assumptions.**

(A1) **Bounded scores.** There exists $B_\pi < \infty$ such that for all histories $(q, o_{<t})$ and tokens $o_t$,
$\big\| \nabla_\theta \log \pi_\theta(o_t \,|\, q, o_{<t}) \big\| \le B_\pi$.

(A2) **Bounded rewards & finite clipping.** For each rollout $o_i$, $R_i \in [0,1]$ and $r_{\mathrm{seq},i}(\theta) \in [\,1 - \varepsilon_{\mathrm{low}},\, 1 + \varepsilon_{\mathrm{high}}\,]$ with $0 < \varepsilon_{\mathrm{low}}, \varepsilon_{\mathrm{high}} < \infty$.

(A3) **Non-degenerate batch dispersion.** On accepted groups, $\mathrm{Var}(\{R_i\}_{i=1}^G) \geq \tau_{\mathrm{var}} > 0$, so $\hat{A}_i = (R_i - \bar{R})/\mathrm{std}(R)$ are well-defined.

(A4) **Vanishing filtering.** $\tau_{\mathrm{adv}}^{(t)} \downarrow 0$ and the law of $\hat{A}_i$ has a continuous density at 0, so $\mathbb{P}(|\hat{A}_i| \leq \tau_{\mathrm{adv}}^{(t)}) \to 0$.

(A5) **Stepsizes.** Robbins–Monro conditions: $\sum_t \eta_t = \infty$ and $\sum_t \eta_t^2 < \infty$.

**Lemma 6 (Bounds on per-sample gradients and second moments).** Let $o = (o_1, \ldots, o_{|o|})$ and $r_{\mathrm{seq}}(\theta)$ denote the (clipped) sequence ratio. Then

$$\nabla_\theta r_{\mathrm{seq}}(\theta) \;=\; r_{\mathrm{seq}}(\theta) \frac{1}{|o|} \sum_{t=1}^{|o|} \nabla_\theta \log \pi_\theta(o_t \,|\, q, o_{<t}), \qquad \left\| \nabla_\theta r_{\mathrm{seq}}(\theta) \right\| \leq (1 + \varepsilon_{\mathrm{high}}) B_\pi. \quad (52)$$

Moreover, the PPO-style term is piecewise smooth and its gradient magnitude is bounded by $C_1 := (1 + \varepsilon_{\mathrm{high}}) B_\pi |\hat{A}|$; together with (A3), $|\hat{A}| \leq \frac{1}{\sqrt{\tau_{\mathrm{var}}}}$ yields a uniform second-moment bound $\mathbb{E}[\|\nabla_\theta \ell_i(\theta)\|^2] \leq C_2 < \infty$.

**Lemma 7 (Asymptotic unbiasedness under vanishing filtering).** Let $g(\theta)$ denote the full (unfiltered) stochastic gradient and $\tilde{g}_\tau(\theta) = \frac{1}{G} \sum_{i:\, |\hat{A}_i| > \tau} g_i(\theta)$ the filtered version with $\frac{1}{G}$ normalization. Under (A4) and the bounded second moments above,

$$\lim_{\tau \downarrow 0} \left\| \mathbb{E}[\tilde{g}_\tau(\theta)] - \mathbb{E}[g(\theta)] \right\| \;=\; 0 \quad \text{for all } \theta. \tag{53}$$

**Theorem 1 (Convergence to a stationary point of the surrogate).** Suppose (A1)–(A5) hold. Then the iterates of stochastic gradient ascent on $\mathcal{J}_{\mathrm{RL}}(\theta)$ with the dynamic filtering scheme converge almost surely to the set of stationary points of the surrogate objective.

*Proof sketch.* By Lemma 6 and the reward boundedness (Lemma 3), the stochastic gradients have uniformly bounded second moments; the objective is bounded and piecewise smooth (kinks of measure zero). Lemma 7 guarantees that the bias due to filtering vanishes as $\tau_{\mathrm{adv}}^{(t)} \to 0$. Therefore the noisy gradient process forms a Robbins–Monro stochastic approximation with asymptotically unbiased gradients and square-summable noise, yielding a.s. convergence to stationary points of $\mathcal{J}_{\mathrm{RL}}$ (e.g., Kushner–Yin/Bottou).

**Remarks.** (i) The min-with-clipping introduces bias w.r.t. the *true* off-policy objective, but ensures variance control and stability; the theorem concerns the surrogate we optimize. (ii) Sequence-level ratios and sequence-level clipping align the gradient scale with the sequence reward, avoiding token/sequence granularity mismatch and contributing to the boundedness needed above. (iii) In practice, we keep $\tau_{\mathrm{var}}$ and the clip window fixed and decay $\tau_{\mathrm{adv}}$, which satisfies the lemmas' conditions and matches our training protocol.

