# OpenReview forum: "Failure makes the agent stronger: Enhancing Accuracy through Structured Reflection for Reliable Tool Interactions"
_ICLR.cc/2026/Conference — ICLR 2026 Conference Withdrawn Submission_

### Official Review · Reviewer_p6We · 2025-10-30

**Soundness:** 1
**Presentation:** 1
**Contribution:** 2
**Rating:** 2
**Confidence:** 4

**Summary:**

This paper adopts reinforcement learning (such as GRPO) to incentivize the reflective capabilities of large language models (LLMs). It first identifies four types of mistakes that frequently occur during LLMs' task-solving process. Based on these mistake types, the author proposes an automatic data synthesis approach to generate a prefix that includes predefined mistakes, forcing LLMs to reflect on the context and generate a new action. Experiments on several datasets validate the effectiveness of the proposed method.

**Strengths:**

1. Enabling "reflection ability" of LLMs is an urgent and important topic for tool-use agents. This paper builds a new dataset, Tool-Reflection-Bench, and uses reinforcement learning (e.g., GRPO) to incentivize LLMs' reflection abilities.

2. The data synthesis approach in this work is well-organized and easy to follow. This simple yet efficient approach can automatically generate reflection-oriented training data (trajectories and prefixes).

**Weaknesses:**

1. I am not clear about the training data. Does each example in the training data contain a prefix $x$, which is then fed into LLMs for continuous action generation until reaching the final answer?

2. What is the main difference between existing papers like [1], especially considering that "reflection" has been a common technique? It seems like the main contribution is enabling reflection using RL rather than supervised fine-tuning.

3. Experiments in this work lack the necessary comparison with existing baselines, such as ToolLLama and ToolACE, other advanced baselines in the BFCL leaderboard.

4. The main figure of this paper should be polished. The current version is not clear. I suggest that the author highlight the main methodology design and the differences with existing work.

5. I am a bit confused about the experiment setup. Does the author put all candidate tools into the context of LLMs, and then prompt the LLMs to call appropriate tools on demand? If so, what if the tool scale is large and the tool description exceeds the model context length?

---
### Reference

[1] Advancing Tool-Augmented Large Language Models via Meta-Verification and Reflection Learning

[2] ToolACE: Winning the Points of LLM Function Calling

**Questions:**

See above weakness.

---

### Official Review · Reviewer_CgsU · 2025-11-01

**Soundness:** 3
**Presentation:** 3
**Contribution:** 2
**Rating:** 4
**Confidence:** 3

**Summary:**

This paper proposes a structured reflection method to improve tool-calling capabilities of large language models through explicit error diagnosis and correction. The authors introduce Tool-Reflection-Bench, a benchmark constructed by systematically perturbing correct tool calls to create failure cases, and design a multi-dimensional reward mechanism for reinforcement learning. The method combines DAPO and GSPO objectives to train models to reflect on failed tool calls and generate corrected ones. Experiments on BFCL v3 and Tool-Reflection-Bench demonstrate improvements in multi-turn tool-call accuracy and error recovery rates.

**Strengths:**

- The paper addresses a genuine problem in tool-augmented LLMs by making error recovery an explicit, trainable capability rather than relying on heuristic prompting, which represents a meaningful shift from prior self-correction approaches.

- The reward design is well-motivated with multiple components (format, tool name, parameters, semantic consistency) that provide granular signals for tool-calling scenarios.

- The experimental results show consistent improvements across multiple base models on both benchmarks, with particularly strong gains in repair rates that surpass closed-source models of similar scale.

**Weaknesses:**

- The benchmark construction methodology raises concerns about generalization, as all four perturbation types (call-order swap, redundant call, missing call, argument error) are synthetic and may not capture the full distribution of real-world tool-calling failures that occur naturally in deployed systems.

- The paper lacks critical ablation studies to validate design choices—it seems there is no analysis of individual reward components, no comparison between DAPO+GSPO versus alternatives (standard PPO, pure DAPO, pure GSPO), and no investigation of how much improvement comes from the reflection mechanism versus the enhanced reward structure.

**Questions:**

How does your method perform on naturally-occurring tool-calling errors from real user interactions or other benchmarks, and what percentage of real-world failures fall into your four perturbation categories versus other failure modes not covered by your taxonomy?

---

### Official Review · Reviewer_rfh8 · 2025-11-01

**Soundness:** 3
**Presentation:** 3
**Contribution:** 2
**Rating:** 4
**Confidence:** 4

**Summary:**

This paper introduces structured reflection in which transforming the "from error to repair" process into a trainable action, to solve the problem that models  often fail to recover from errors and tend to repeat the same mistakes. The authors combine objective functions of DAPO and GSPO in training and desigh a reward  mechanism for tool calling. The authors also introduce Tool-Reflection-Bench for evaluation. The results on BFCL v3 and Tool-Reflection-Bench show great improvements in multi-turn tool-call success rates and error recovery.

**Strengths:**

1. This work introduces a novelty training mechanism about reflection, making contributions on models' tool-call abilities. And the results show great improvement for models.

2. This work proposes 4 pertubations and constructs Tool-Reflection-Bench for evaluation.

3. The paper is well-written and easy to follow, with enough case studies.

**Weaknesses:**

1. Limited Discussion on generalization and model scale: Although  the paper discusses about performance on 4 perturbations and BFCL v3, it still lacks analysis about how well the learned capability generalizes to more failures in real world. Also, the models in experiments are 4B to 8B. The results remains unclear on much larger models.

2. Comparison with other RL methods : The paper chooses a complex RL pipeline with DAPO+GSPO and increase the complexity and cost without doubt. However, the paper lacks a discussion on the cost-benefit trade-off with other traditional RL methods.

3. Failure of reflection itself: This paper focuses on using reflection to fix tool-call errors. However, the failure of reflection itself also makes sense. I think authors should also pay attention to the error reflection or the inconsistency between reflection and execution in the process.

**Questions:**

1. How does the learned reflection capability generalize to failures beyond the four specific perturbation types? Additionally, how do you expect the method's effectiveness to scale with much larger models (e.g., 70B+)?

2. Could you analysis the trade-off between the complexity of the RL pipeline and the performance gains, particularly when compared to simpler, less costly alternatives?

3. Have you analyzed instances of "reflection failure," where the model generates an incorrect reflection or its subsequent action is inconsistent with its own reflection?

---

### Official Review · Reviewer_Ni9n · 2025-11-01

**Soundness:** 3
**Presentation:** 3
**Contribution:** 2
**Rating:** 4
**Confidence:** 3

**Summary:**

This paper aims to "transform self-correction [by LLMs] into a trainable and controllable capability" (line 186).
The basic idea is to convert self-correction into a supervised training problem x ↦ y.  Each synthetic training example starts with a correct y and synthesizes a plausible erroneous version x that must be corrected into y (lines 100-102, 112-113).

(Cf. a denoising autoencoder, where a supervised model is trained to undo a corruption step, or similarly BART and its ilk.  However, those are merely artificial tasks used for representation learning, whereas this paper is interested in the real task of correcting errors that actually arise during LLM message sequences.  Such real-task settings in prior work include grammatical error correction, e.g., [Felice 2016](https://www.cl.cam.ac.uk/techreports/UCAM-CL-TR-895.pdf), [Kiyono et al. 2019](https://aclanthology.org/D19-1119/), and code correction, e.g., [Gupta et al. 2017](https://ojs.aaai.org/index.php/AAAI/article/view/10742).)

This is really a general recipe for training self-correction, and a sensible idea if not a new one.
The difficulty is in figuring out how to create "plausible" corruptions.

This paper specifically focuses on tool calling.  Some points worth noting:

1. It uses a few simple hand-designed corruption operators (lines 220-227) based on manual analysis of error patterns (line 98).

1. The corruptions are not *learned* to match the actual distribution of errors, as I had imagined at first.

1. While the corrections are observed, the chains of thought needed to achieve them ("reflections") are provided by humans.

1. The method is trained by RL and not just by SFT, which introduces more complexity since a finicky loss function must be specified (section 3.2).

**Strengths:**

Quality: The basic idea makes sense and the execution seems competent.  I agree that self-correction by LLMs should be trainable, and that supervised trajectories can be modified to provide examples that need to be corrected back to the supervised trajectory.

Significance: The improvements on the existing BFCL benchmark (Table 1) seem quite strong -- although I am not familiar with the benchmark and may not be interpreting the results appropriately.

The approach might not be too hard to adapt to training some other kinds of LLM self-correction (this paper just focuses on tool calls).

Clarity: I was mostly able to follow the paper, although I have some confusions (see questions below).  Detailed appendices are provided; the "case studies" in section A.3 are helpful for making the method more concrete.

**Weaknesses:**

The general direction is not exactly new.  The detailed method is somewhat complicated, and is not compared to simpler alternatives.

The results are presented very briefly, and are not really discussed in the text (just recapitulated from the tables).  I don't have a clear understanding of the qualitative patterns of the results.

There may have been some missed opportunities.  See questions below.

**Questions:**

**Related work.** The paper seems at least loosely related to [Wang et al. (2024)](https://aclanthology.org/2024.acl-long.570/), [Xu et al. (2024)](https://arxiv.org/abs/2406.17465), and [Qu et al. (2025)](https://arxiv.org/abs/2410.08197), none of which are cited.  However, those papers actually try executions, and I think this paper doesn't.  Can you discuss the submitted work in the context of that line of prior work?

**Reflection generation.** The prompt at line 691 doesn't seem able to generate a good context-dependent reflection+correction, because it doesn't give any documentation for the tool API or any context about what this particular call was trying to achieve!  Can you explain?

**Cost of human supervision.** Human supervision is used to correct (r,c) to (r*,c*) at (8).  Who provided the supervision?  How expensive was it?  Apparently (line 415) you produced at least 5k examples, though I'm not sure whether that means steps or multi-step trajectories.

**STaR workflow.** The point of this is to get a good reflection r*, since c* is known already.  But after obtaining a few human examples of this to illustrate the style (e.g., for few-shot prompting), perhaps one could use STaR ([Zelikman et al. 2022](https://arxiv.org/abs/2203.14465)) to have the LLM itself fill in r* (given c*) that is achievable by the model and tends to lead to c*?

**Imitation learning.**  Since supervised trajectories $m_0,m_1,\dots$ are available to the learner, why not try imitation learning?  For example, instead of generating errors by artificial corruption, why not train to correct the learner's *currently predicted* action $\tilde{m}\_i$ in the context $m_0,\dots,m_{i-1}$ to the supervised action $m\_i$?

* Apologies that my notation here is a little sloppy.  The currently predicted action may be the result of preceding predictions that are self-corrected; here I am imagining that those preceding predictions and their reasoning steps fall between $m_{i-1}$ and $\tilde{m}\_i$.  If the currently predicted action is wrong, then we want to learn an additional self-correction action, otherwise an "accept" action.

* This standard supervised imitation learning method is imperfect because of exposure bias: the distribution of contexts $m_0,\dots,m_{i-1}$ at test time is produced by the learned policy and probably won't match the training distribution ([Ross et al. 2010](https://arxiv.org/abs/1011.0686)).  But it may still be better than the method proposed in the submission.

* There are certainly *non*-sequential settings where a post-correction model is learned to correct system output to the reference output.  The machine translation community has a whole shared task series around "automatic post-editing" (APE).  Here, exposure bias is not an issue.

**Supervised training.** How important is it to train with the reward of section 3.2 rather than just log-loss (SFT to match the demonstrated (r*, c*) pair)?

**Task reward.** Do you only have rewards from (24)-(25), which ask about matching the demonstration?  (That's what line 419 implies.)  Would it make sense to also reward trajectories where the tool calls manage to achieve the goal in some other way?

**Reward design.** Can you give an example of (11)?  I'm not sure I understand what C\_calls and G\_calls look like.  Are they each a sequence of revisions to a call, with c\_final or g\_final being the final accepted revision in that sequence?  But if so, I have several confusions:
* should the ground truth have $j=0$ in almost all cases?
* such a sequence would be ordered, but you say "multiset," which implies that they are unordered (and presumably ordered arbitrarily in the prompt).
* why doesn't each $c_i$ or $g_j$ have its own reflection
* if c\_ref and g\_ref are reflections, why isn't their main symbol $r$? (perhaps these should be called $r$ and $r*$ as in the previous section?)

**Learning how many times to revise.** Section 3.1.3 creates supervised examples of tool calls that need to be corrected, but don't you also need supervised examples of tool calls that _don't_ need correction (hence, 0 corrections)?  Also, does your system learn how to iteratively correct until convergence?  I'm not sure.

**Long-context recovery.** Mentioned at line 443: what do you mean?

**Table 1 results.** Table 1 evaluates on BFCL, but almost nothing is said about this setup.  What is BFCL?  How many turns are there per trajectory?  What metric is being reported in Table 1, and is it per-turn or per-trajectory?  Do the columns correspond to error categories that are provided by the BFCL dataset?  Are the improvements statistically significant?  Can you say more about whether corrections are applied at the right times (precision/recall), and whether they are the correct type of correction and whether they are successful?  If they are unsuccessful, is a further correction then applied?

**Table 2 results.** This is the test split of your own synthetic dataset, right?  Since your model was trained on the training split of the same dataset, you would expect it to do well, right?  So is this table just a sanity check that your training worked?

---

### Note · Authors · 2025-11-21

I have read and agree with the venue's withdrawal policy on behalf of myself and my co-authors.